# The cost and cost implications of implementing the integrated chronic disease management model in South Africa

Limakatso Lebina[1,2]*, Mary Kawonga[3], Tolu Oni[2,4], Hae-Young Kim[5], Olufunke A. Alaba[6]

**1** Perinatal HIV Research Unit (PHRU), SA MRC Soweto Matlosana Collaborating Centre for HIV/AIDS and TB, Faculty of Health Sciences, University of the Witwatersrand, Johannesburg, South Africa, **2** School of Public Health and Family Medicine, University of Cape Town, Cape Town, South Africa, **3** Department of Community Health, School of Public Health, Faculty of Health Sciences, University of the Witwatersrand, Johannesburg, South Africa, **4** MRC Epidemiology Unit, University of Cambridge, Cambridge, United Kingdom, **5** Department of Population Health, New York University Grossman School of Medicine, New York, New York, United States America, **6** Health Economics Unit, School of Public Health and Family Medicine, University of Cape Town, Cape Town, South Africa

* lebinal@phru.co.za

**Data Availability Statement:** The data collected for this study are available at Lebina, Limakatso (2020): ICDM Model Costs. figshare. Dataset. https://doi.org/10.6084/m9.figshare.11987271.v1

## Abstract

### Background

A cost analysis of implementation of interventions informs budgeting and economic evaluations.

### Objective

To estimate the cost of implementing the integrated chronic disease management (ICDM) model in primary healthcare (PHC) clinics in South Africa.

### Methods

Cost data from the provider's perspective were collected in 2019 from four PHC clinics with comparable patient caseloads (except for one). We estimated the costs of implementing the ICDM model current activities for three (facility reorganization, clinical supportive management and assisted self-management) components and additional costs of implementing with enhanced fidelity. Costs were estimated based on budget reviews, interviews with management teams, and other published data. The standard of care activities such as medication were not included in the costing. One-way sensitivity analyses were carried out for key parameters by varying patient caseloads, required infrastructure and staff. Annual ICDM model implementation costs per PHC clinic and per patient per visit are presented in 2019 US dollars.

### Results

The overall mean annual cost of implementing the ICDM model was $148 446.00 (SD: $65 125.00) per clinic. Current ICDM model activities cost accounted for 84% ($124 345.00) of

**Funding:** LL received funding from the South African Medical Research Council (SA MRC) Self-Initiated Research Grant (ID:494184). The views and opinions expressed are those of the author(s) and do not necessarily represent the official views of the SA MRC. The funders had no role in study design, data collection and analysis, decision to publish, or preparation of the manuscript.

**Competing interests:** The authors have declared that no competing interests exist.

the annual mean cost, while additional costs for higher fidelity were 16% ($24 102.00). The mean cost per patient per visit was $6.00 (SD:$0.77); $4.94 (SD:0.70) for current cost and $1.06 (SD:0.33) for additional cost to enhance ICDM model fidelity. For the additional cost, 49% was for facility reorganization, 31% for adherence clubs and 20% for training of nursing staff. In the sensitivity analyses, the major cost drivers were the proportion of effort of assisted self-management staff and the number of patients with chronic diseases receiving care at the clinic.

## Conclusion

Minimal additional cost are required to implement the ICDM model with higher fidelity. Further research on the cost-effectiveness of the ICDM model in middle-income countries is required.

## Background

Chronic diseases are a major public health challenge, accounting for 60% of all deaths, with 35% and 40% of deaths due to chronic diseases occurring in low and middle income countries respectively [1–3]. Chronic diseases can cost up to 7% of a country's gross domestic product (GDP) due to the undesirable effect these diseases have on economic activities and increased public health and social welfare expenditure [3, 4]. In high-income settings such as the USA, Europe and Spain, the cost of the increasing number of chronic illnesses account for 75%, 80% and 77% of the total healthcare cost respectively [5, 6]. The key drivers of cost in health systems are increased utilization of services, medication and health system adaptation of service delivery [3].

South Africa, like many low and middle-income countries, has a dual burden of chronic diseases, with a high prevalence of both communicable and non-communicable chronic diseases [7, 8]. Data from Free State Province indicate that 24% of the population suffer from chronic diseases [9]. Results from a Cape Town study showed that prevalence of multi-morbidity is approximately 23%, and chronic diseases account for 45% of all primary healthcare (PHC) consultations [10]. Nevertheless, the diagnostic tools, training and supervision of clinical staff as they manage and provide care for chronic conditions are inadequate, particularly at the PHC level [11, 12]. In an effort to address this challenge, the South African Department of Health developed and implemented the integrated chronic disease management (ICDM) model [13]. The ICDM model was developed based on the principles of the chronic care model (CCM) and innovative care for chronic conditions (ICCC) framework [13, 14]. Research in other settings has indicated that integrated chronic care models improve patient care and health outcomes [15–17].

The aim of the ICDM model is to provide a comprehensive chronic disease management model that reduces healthcare utilization and promotes self-management among patients with chronic diseases [13, 14]. Patients who are incorporated into the ICDM model include adults and children with chronic communicable (HIV/AIDS and tuberculosis) and non-communicable (hypertension, diabetes, asthma, chronic obstructive pulmonary disease, mental health and epilepsy) diseases [13]. The ICDM model consists of four inter-related components, namely facility reorganization, clinical supportive management, assisted self-management and strengthening of support systems [13, 14].

Facility reorganization activities include management of patient flow, bookings and records to enhance clinic operational effeciency [13]. Clinical supportive management includes the activities of the district clinical specialist team (DCST) and the training of nurses on primary healthcare and management of the conditions included in the ICDM model [13]. Assisted self-management activities aim to empower patients to become involved in their disease management and be supported at community level [13]. Community level support is provided by ward-based outreach teams (WBOTs) and community healthcare workers (CHCW). Patients with chronic conditions who are stable on treatment are offered the option of collecting pre-packed medication at the PHC clinic or at other outlets under the central chronic medicine dispensing and distribution (CCMDD) programme [13]. The strengthening of support systems promotes collaborations between the clinic and other departments like school health and support structures for health services delivery such as community organizations [13]. An assessment of the ICDM model implementation has shown enhancements in patients' records, improved health outcomes for HIV-positive patients on antiretroviral medication, but no substantial improvements for patients with hypertension [18, 19].

Developing, implementing and evaluating effectiveness and implementation outcomes are essential in public healthcare research [20]. The implementation outcomes or measures of successful implementation an intervention or a programme, include amongst others, fidelity, implementation costs, acceptability and sustainability [21]. Implementation costs of interventions reveals the feasibility, scalability and sustainability of proposed integrated care interventions [20]. Implementation costs is one of the implementation outcomes that also allows decision makers to determine and choose which interventions are efficient and equitable [20]. Fidelity is the degree to which the implementation of a programme follows the original design as outlined in the guidelines [22, 23]. A process evaluation of the implementation fidelity of the ICDM model indicates that the level of fidelity (adherence) to guidelines varies between clinics, with some clinics having high scores (80%-89%), and others having medium (70%-79%) and low (<70%) scores [24]. Chronic care models are also cost-effective as they reduce healthcare services utilization and improved disease management [15–17]. However, there are no data on the cost of implementing the ICDM model in South Africa. The objective of this study was to estimate the empirical implementation cost of each of the components of the ICDM model in two health districts to inform planning, scaling-up and further economic evaluations. In addition, we assessed if the degree of fidelity to the ICDM model guidelines has an impact on the cost of implementing the ICDM model.

## Methods

### Overview

This analysis is nested within a larger study to assess the level and determinants of fidelity in the implementation of the ICDM model in the two health districts in South Africa [25]. We evaluated the cost of implementing three (facility reorganization, clinical supportive management, assisted self-management) of the four components of the ICDM model from the health system perspective. The strengthening of support systems was not included in the cost analysis as the activities are extensive and most of the costs are covered by collaborating organizations. In addition to evaluating the current cost of operating the ICDM model, we estimated the costing of the additional activities or infrastructure required to implement the ICDM model with a high degree of fidelity in a PHC clinic per annum. Due to lack of data on the ICDM model's effectiveness in optimizing clinical outcomes, we conducted a cost analysis using the empirical micro-costing. Standard healthcare service costs like medication, laboratory investigations and management of complications were excluded from the analysis to keep the focus on the ICDM

model activities. The standard costs would be incurred irrespective of the ICDM model and their inclusion would overestimate the costs of the ICDM model activities.

## The setting

The study was conducted in two health districts in South Africa, the West Rand (WR) district in the Gauteng province and the Dr. Kenneth Kaunda (DKK) district in the North West province [25]. These two health districts, as well as Bushbuckridge health district in Mpumalanga, were the pilot sites for implementing the ICDM model in South Africa on the initiative the South African National Department of Health in 2011 [26]. The two health districts were selected for this study as they have comparable population sizes and burden of disease. The population sizes in the two districts are similar (811 000 in WR and 716 000 in DKK health district) [27]. In 2011, the provincial HIV prevalence was also comparable (12.4% and 13.3% in Gauteng and North West, respectively [28]). Hypertension prevalence is high (31%- 39.7%) in both health districts, with high health service utilization for chronic non-communicable diseases [27]. In Gauteng, 63% of the households rely on state health facilities (clinics and hospitals) for health services, while 73% of the households in North West depend on state health facilities [29]. We selected these two districts as the ICDM model had been implemented in these districts for over 7 years. The cost analysis would therefore be representative of medium to long-term scale-up scenarios.

## Ethics approval

The study was approved by the ethics committees of the University of Cape Town (Ref: 127/2018) and the University of the Witwatersrand (Ref: R14/49).

## Data collection

**Clinic selection.**    Four of the 16 PHC clinics that formed part of the larger 'degree of fidelity' study [25] were selected based on their ICDM model implementation fidelity scores and patient caseloads. One of the clinics with the highest [clinics A (87.%) and B (84.8%)] fidelity scores and one of the clinics with the lowest [clinics C (76.6%) and D(65.8%)] fidelity scores [24]. were chosen per health district for the cost analysis. The four clinics had comparable characteristics (personnel and patient consultations) as summarized in Table 1. However, clinic A had more consulting rooms and fewer patients consulting for chronic diseases management.

**Estimation of costs.**    Costs were stated according to the 2019 prices and converted to US dollars (US$) at an exchange rate of ZAR 14.42 equal to US$ 1 [30]. The costs are reported as annual costs per PHC clinic and per patient with chronic disease per visit. The current cost and the additional cost of required infrastructure or unexecuted recommended ICDM model activities were estimated as outlined below.

**Facility reorganization cost.**    The current costs were estimated based on interviews with key personnel, budget reviews and estimating the proportion of resources allocated to patients with chronic disease based on the clinic consultations records. On average, 59% of the patients attending the four PHC clinics were consulting for chronic disease care. Therefore, we used this proportion to apportion facility reorganization costs for the current activities. Items included in the cost estimate for current facility reorganization were building (reception, waiting areas, vital signs station and consulting rooms), equipment and furniture, and building maintenance costs. Additional infrastructure cost for facility reorganization were based on the estimates according to the guidelines, as well as current cost. The ICDM model guidelines recommend that patients with chronic diseases should have separate waiting rooms, a vital signs

**Table 1. Characteristics of the primary healthcare clinics included in the cost analysis of implementing the Integrated Chronic Disease Management (ICDM) model.**

| Variable | Clinic A | Clinic B | Clinic C | Clinic D |
|---|---|---|---|---|
| *Fidelity to the ICDM model* | Score (Percentage) | | | |
| Overall fidelity of the ICDM model implementation | **136/158 (87.3)** | **134/158 (84.8)** | **121/158 (76.6)** | **104/158 (65.8)** |
| Facility reorganization | 29/37 (78.4) | 27/37 (73.0) | 30/37 (81.1) | 29/37 (78.4) |
| Clinical supportive management | 37/39 (94.9) | 33/39 (84.6) | 29/39 (74.4) | 20/39 (51.3) |
| Assisted self-management | 36/39 (92.3) | 37/39 (94.9) | 27/39 (69.2) | 30/39 (76.9) |
| Strengthening of support systems | 34/43 (79.1) | 37/43 (86.1) | 35/43 (81.4) | 25/43 (58.1) |
| | *Number of Personnel* | | | |
| Professional nurses | 4 | 5 | 6 | 10 |
| Enrolled nurses | 3 | 0 | 5 | 1 |
| Medical officers | 1 | 1 | 5 | 1 |
| Pharmacists assistant | 0 | 0 | 1 | 0 |
| Administrative staff | 5 | 8 | 3 | 1 |
| Counsellors | 2 | 6 | 4 | 7 |
| WBOT^ Leaders | 4 | 2 | 4 | 2 |
| WBOT^/CHCW# | 4 | 13 | 50 | 17 |
| Nurse patient ratio | 333 | 731 | 370 | 340 |
| Medical officer patient ratio | 2328 | 3655 | 814 | 3735 |
| *Patient consultations per month. per facility        mean (SD)* | | | | |
| Total primary healthcare headcount | 2328 (150) | 3655 (206) | 4068 (146) | 3735 (245) |
| Total patients >20 years per month per facility | 1530 (126) | 2457 (187) | 2865 (106) | 2795 (233) |
| Total patients with chronic diseases consultations* | 981 (67) | 2653 (133) | 2922 (302) | 1802 (144) |
| Proportion of patients with chronic diseases consultation to total headcount | 42% | 73% | 72% | 48% |
| Adults in care for HIV/AIDS | 687 (38) | 2240 (27) | 2004 (48) | 1434 (13) |
| New TB diagnosis | 2 (1) | 4 (5) | 3 (1) | 14 (3) |
| Diabetic patients consultation | 63 (8) | 90 (24) | 144 (30) | 57 (4) |
| Hypertensive patients consultations | 243 (34) | 344 (119) | 768 (157) | 227 (126) |
| Nurse patient ratio | 259 | 406 | 452 | 414 |
| *Infrastructure* | | | | |
| Total area in m$^2$ | 557 | 1367 | 491 | 442 |
| Number of consulting rooms | 8 | 5 | 2 | 3 |
| Vital signs stations | 1 | 1 | 1 | 1 |
| Waiting areas | 2 | 1 | 1 | 2 |
| Reception/Medical Reocords | 1 | 1 | 1 | 1 |

^WBOT–Ward-based outreach teams

#CHCW–Community healthcare worker

*Based on the number of patients with each disease. and not taking into consideration multi-morbidity

station and consultation rooms. On evaluation of the fidelity and patient flow analysis, some clinics did not meet the guidelines for adequate infrastructure. Therefore, the additional facility reorganization cost was calculated by calculating the cost of building an additional three consultation rooms for clinics C and D only, as clinics A and B had adequate consultation rooms according to the guidelines. Furthermore, additional facility reorganization costs included building a waiting area, a vital signs station, a CCMDD kiosk and a multi-purpose meeting room for adherence clubs meetings for each of the four clinics. The South African building cost of US$ 561.17 per m$^2$ was used to estimate the cost of building the additional

recommended areas based on valued office building costs [31]. The size of the rooms, furniture and equipment requirements were based on the data collected and the guideline recommendations [32, 33]. The cost of the equipment and furniture were sourced from large furniture and medical supplies stores. A discount rate of 7% [34] was applied to calculate the annual cost of buildings, furniture and equipment with an estimated life span of 20 years, 10 years and 5 years respectively.

**Clinical supportive management cost.** The current cost of clinical supportive management were calculated based on costs of the DCST providing support to each PHC in the district. The DCST in the DKK district did not provide support to PHC clinics on management of patients with chronic diseases, so the cost of providing the DCST in the WR district was used for the DKK district. The proportion of time apportioned to patients with chronic diseases was determined by interviewing the DCST leader in the WR health district. The proportion allocation of personnel (family physician, senior pediatric medical officer, pediatric nurse and senior PHC nurse) costs and telecommunication costs for the DCST to ICDM model was calculated and divided by number of clinics in each health district. The additional cost of clinical supportive management was for training nurses on PHC- and nurse-initiated management of antiretroviral treatment (NIMART) [13]. The training cost was obtained from a training organization that delivers the two training courses on behalf of the Department of Health. We worked on the assumption that all professional nurses at the clinics would have to be trained to account for staff turn-over. Travel cost was not included as the training is usually delivered in the district.

**Assisted self-management cost.** The current costs of CHCW, WBOT and supervision was estimated according the current salaries and the proportion of time allocated by the staff members to support patients with chronic diseases. Staff salary values were informed by the National Department of Health salary scales. All four of the PHC clinics had no functional adherence clubs, thus the additional cost of training staff on adherence support and ongoing provision of the adherence clubs was sourced from the literature [35]. Data on adherence clubs' cost were only available for 2011 [35], and the average inflation of 5.4% was applied to estimate the costs for 2019.

**Sensitivity analysis.** One-way sensitivity analyses were performed on all cost estimates that were major (> 4%) components of the cost for both current and additional costs. We varied the discount rate from 4% to 9% (based on data for the last 5 years [34]) to assess the different scenarios of cost for current and additional building costs. In addition, under the current cost, building and building maintenance were varied for the proportion of patients that consult for chronic disease using one standard deviation of the mean of 59% as this was applied in the initial analysis. For the additional costs, building cost was varied depending on whether additional consulting rooms are required. One standard deviation (SD) was applied to calculate the highest and lowest cost for WBOTs and WBOT supervision from the mean effort of 62% and 52% respectively. The mean number of patients and the SD were used in the sensitivity analyses for the adherence club cost. The SD of the numbers of nurses that have to be trained was applied for the NIMART and PHC training sensitivity analysis.

## Results

### Implementation cost of the ICDM model

**The overall ICDM model implementation cost.** Based on the data collected from the four PHC clinics, the annual cost (current and additional costs) of implementing the ICDM model per PHC clinic varied from $77 726.00 in clinic A to $232 103.00 in clinic C (Table 2). The mean overall annual cost of implementing the recommended activities of the ICDM

**Table 2. The estimated annual current and additional costs to enhance the fidelity to the ICDM model recommended activities in the four study clinics–A and B high fidelity, C moderate and D low fidelity.**

| | ICDM Model Component | Items | Clinic A | Clinic B | Clinic C | Clinic D |
|---|---|---|---|---|---|---|
| Current Costs | Facility reorganization | Building | $12 432.88 | $52 559.23 | $18 681.49 | $11 321.36 |
| | | Furniture | $1 580.38 | $2 162.55 | $1 726.56 | $1 346.59 |
| | | Equipment | $746.67 | $ 867.02 | $443.22 | $390.56 |
| | | Building maintenance | $ 3 118.68 | $13 184.02 | $4 686.08 | $2 839.86 |
| | Clinical supportive management | DCST | $1 019.93 | $1 218.25 | $1 019.93 | $1 218.25 |
| | Assisted self-management | WBOT Supervision | $31 128.78 | $29 288.06 | $31 128.78 | $29 288.06 |
| | | WBOT | $ 11 618.26 | $37 759.34 | $145 228.22 | $49 377.59 |
| | Total current costs | | **$61 645.58** | **$137 038.46** | **$202 914.28** | **$ 95 782.28** |
| Additional Costs to enhance Fidelity | Facility reorganization | Building | $ 7 395.30 | $7 395.30 | $9 296.95 | $9 296.95 |
| | | Furniture | $150.46 | $150.46 | $1 630.84 | $1 630.84 |
| | | Equipment | $56.45 | $56.45 | $807.28 | $ 807.28 |
| | | Building maintenance | $1 855.05 | $1 855.05 | $2 332.06 | $2 332.06 |
| | Clinical supportive management | NIMART Training | $1 715.08 | $ 2 143.85 | $2 572.61 | $ 4 287.69 |
| | | PHC Training | $1 400.83 | $1 751.04 | $2 101.24 | $3 502.07 |
| | Assisted self-management | Adherence Club | $3 507.47 | $9 485.54 | $10 447.32 | $6 442.87 |
| | Total additional costs | | **$16 081** | **$22 838** | **$29 188** | **$28 300** |
| Estimated total costs of implementing ICDM model in each PHC clinic (Current costs + additional costs to enhance fidelity) | | | **$77 726.21** | **$159 876.14** | **$232 102.59** | **$124 082.04** |
| Costs per patient per visit | | | | | | |
| Current costs per patient per visit | | | $5.24 | $4.30 | $5.79 | $4.43 |
| Additional costs per patient per visit | | | $1.37 | $0.72 | $0.83 | $1.31 |
| Total mean costs per patient per visit | | | $6.60 | $5.02 | $6.62 | $5.74 |

model per facility was $148 447.00 (SD: $65 125.00), and the mean cost per patient per visit was $6.00 (SD:$0.77). Capital costs were 24% ($35 760.09) of the total annual mean costs. Current cost contributed 84% ($124 345.00) of the total annual mean cost, while additional cost accounted for 16% ($24 102.00) (Table 3). Almost two-thirds of the current cost came from the personnel cost of the WBOTs/CHCW and their supervision under assisted self-management (66%; $91 204.00). For the additional annual cost, facility reorganization accounted for 49% ($9 668.64), while adherence clubs and the training cost for nurses were 31% ($7 470.80) and 20% ($4 868.60) respectively. The additional recurrent costs to achieve higher ICDM model fidelity per patient per visit was $0.62 (SD: 0.15).

In all of the four PHC clinics, the current cost was higher than the additional (infrastructure, training and adherence clubs) cost required to enhance the degree of fidelity. The cost per patient per visit for each of the four clinics for the current cost ranged from $4.30 to $5.79 (mean: $4.94; SD:0.70) and the additional cost from $0.72 to $1.37 (mean: $1.06; SD:0.33). The clinics with the higher level of ICDM model implementation fidelity had a lower additional annual cost [clinic A ($16,081) and clinic B($22,838)], compared to those with lower degree of fidelity [clinic C ($29,188) and clinic D (28,299)].

**Facility reorganization.** The mean cost of facility reorganization contributed 25.8% ($32 021.79) to the current annual cost and 49% ($9 668.64) to the additional annual cost. The building costs for clinic B are higher compared to the other clinics as it is based in a large repurposed manucipality building. These additional facility reorganization cost included the capital investment of building additional facilities dedicated for chronic patients, such as a vital signs station, a waiting area, a CCMDD kiosk and a multi-purpose room in all four clinics, as well as three consulting rooms in two of the clinics. The cost of maintaining the

**Table 3. Mean current and additional annual costs of implementing the recommended activities for ICDM model per PHC clinic.**

|  | ICDM Model Component | Items | Estimated annual costs Mean (SD) | Proportion of costs |
|---|---|---|---|---|
| Current Costs | Facility reorganization | Building | $23 748.74 (19 478) | 19.1% |
|  |  | Furniture | $1 704.02 (343) | 1.4% |
|  |  | Equipment | $611.87 (231) | 0.5% |
|  |  | Building maintenance | $5 957.16 (4 886) | 4.8% |
|  | Clinical supportive management | DCST | $1 119.09 (115) | 0.9% |
|  | Assisted self-management | WBOT Supervision | $30 208.42 (1 063) | 24.3% |
|  |  | WBOT | $60 995.85 (58 333) | 49.1% |
|  | *Total current costs* |  | **$124 345.15 (60 776)** | 100% |
| Additional Costs | Facility reorganization | Building | $8 346.12 (1 098) | 34.6% |
|  |  | Furniture | $890.65 (855) | 3.7% |
|  |  | Equipment | $431.87 (433) | 1.8% |
|  |  | Building maintenance | $2 093.55 (275) | 8.7% |
|  | Clinical supportive management | NIMART Training | $2 679.81 (1 128) | 11.1% |
|  |  | PHC Training | $2 188.80 (928) | 9.1% |
|  | Assisted self-management | Adherence Club | $7 470.80 (3 146) | 31.0% |
|  | *Total additional costs* |  | **$24 101.60 (6 040)** | 100% |
| Current costs |  |  | $124 345.15 | 85% |
| Additional costs |  |  | $24 101.60 | 15% |
| Total mean costs of implementing ICDM model |  |  | **$148 446.75 (65 125)** | 100% |

additional spaces, for instance ensuring cleanliness, water and sanitation was estimated at $2 093.55 per annum. The mean cost of equipment and furniture was estimated at $1 322.52 per annum.

**Clinical supportive management.** Clinical supportive management activities accounted for 0.9% ($1 119.09) of the total current annual cost of implementing the ICDM model. The mean cost of the DCST providing support to each health facility was low as the proportion of time allocated to support each clinic was low. The DCST cost varied slightly between the two health districts; the cost in WR was $1 022.76 while in the DKK it was $1 221.63. The cost for providing NIMART and PHC training for the nurses was significantly higher and accounted for 20% ($4 868.60) of the additional annual cost.

**Assisted self-management.** Interviews with managers and a review of the reports of the CHCW and WBOT staff members indicated that WBOTs/CHCW allocate 62% (SD: 43%) of their time to adherence support and tracing defaulters on chronic medication. The mean number of WBOT/CHCW per facility was 21 (SD:20) (Table 1), and each earns $242 72 per month. The WBOTs and CHCW are supervised by nurses for a portion of their time at a mean of mean 52% (SD:12%). In two of the PHC clinics, two professional nurses provided supervision (clinic A and C) clinics and in the other two, four enrolled nurses provided this service (clinic B and D). The estimated current mean annual cost of providing assisted self-management was $60 995.85 (SD:$58,33) for WBOTs/CHCW and $30 208.42 (SD:1,063) for their supervision. The costs per clinic for WBOTs/CHCW differed by clinic as it is greatly affected by the number of WBOTs/CHCW based at each clinic according to the number of community wards. For example the cost for clinic C was $145 228.22 for 50 WBOTs/CHCW versus $11 618.25 in clinic A for 4 WBOTs/CHCW. The additional cost of providing adherence clubs for assisted self-management varied from $3 507.47 in a clinic with 981 patients with chronic disease consultations per month (Clinic A) to $10 477.32 in a clinic with almost 3000 patients per month (Clinic C).

**Sensitivity analysis.** The key parameters in the cost model and the one-way sensitivity analyses are outlined in Table 4 and Fig 1. The current building cost was $18 513.18 at discount rate from 4% and $25 625.80 at 9%; while additional building cost was $6 506.17 and $9 005.79 for 4% and 9% discount rates. Based on the one-way sensitivity analysis, the major cost drivers are the proportion of effort of the WBOTs/CHCW per WBOTs supervisor and the number of patients accessing chronic disease care. Varying the building cost according to whether or not additional consulting rooms are included also revealed that building is a major cost driver in the additional cost (Fig 1B).

## Discussion

From the provider's perspective, we estimated that the annual mean cost of implementing the ICDM model activities are $148 446.75 per clinic or $6.00 per patient with chronic disease per visit. The current cost was the largest component of the overall ICDM model implementation cost. The additional costs were lower for clinics with a higher degree of implementation fidelity. Facility reorganization accounted for 49%, adherence clubs 31% and training of nursing staff 20% of the additional mean cost. Assisted self-management was the most costly component of the ICDM model to implement, and it contributed 73% of the current cost and 31% of the additional cost. The overall ICDM model implementation cost varied between the four study PHC clinics. The major cost drivers were the number of patients accessing services for chronic disease management and associated WBOTs to support assisted self-management and the required additional infrastructure.

The annual cost of implementing the ICDM model activities was $148,446.75 per clinic. The cost of implementing a team-based chronic care model in another study conducted in northern California, USA, was estimated at $2 304 787.00 over 29 months ($79 475.41 per month) [36]. Chronic disease management, particularly in the context of multimorbidity, is the largest expense for health systems [4–6]. More research and strategies to improve the effectiveness of the ICDM model to enhance health outcomes for all chronic diseases are important to support such an expenditure [18, 19]. A systematic review showed other types of integrated care for chronic diseases, based on similar principles as the ICDM model to be cost-effective by reducing health utilization and improving health outcomes [16]. All the studies that were reviewed had been conducted in high-income countries [16]. In this study, the current cost was the largest component of the total estimated mean annual cost of the ICDM model. This

**Table 4. The range of key parameters in the costs model.**

| Key cost parameter | Baseline *(mean)* | Standard deviation | Range |
|---|---|---|---|
| **Current costs** | | | |
| Building–proportion of patients with chronic diseases | 59% | 16% | 43% -75% |
| Building maintenance–proportion of patients with chronic diseases | 59% | 16% | 43% -75% |
| WBOT supervison–effort on ICDM model activities | 52% | 12% | 40%– 64% |
| WBOT–effort on ICDM model activities | 62% | 43% | 18% - 100% |
| **Additional costs** | | | |
| Building–including or excluding additional consulting rooms | 3 | N/A | 0–3 |
| Adherance club–number of patients with chronic diseases | 2 090 | 880 | 1 210–2 970 |
| NIMART training–number of nurses to be trained per annum | 6 | 3 | 3–9 |
| PHC training—number of nurses to be trained per annum | 6 | 3 | 3–9 |

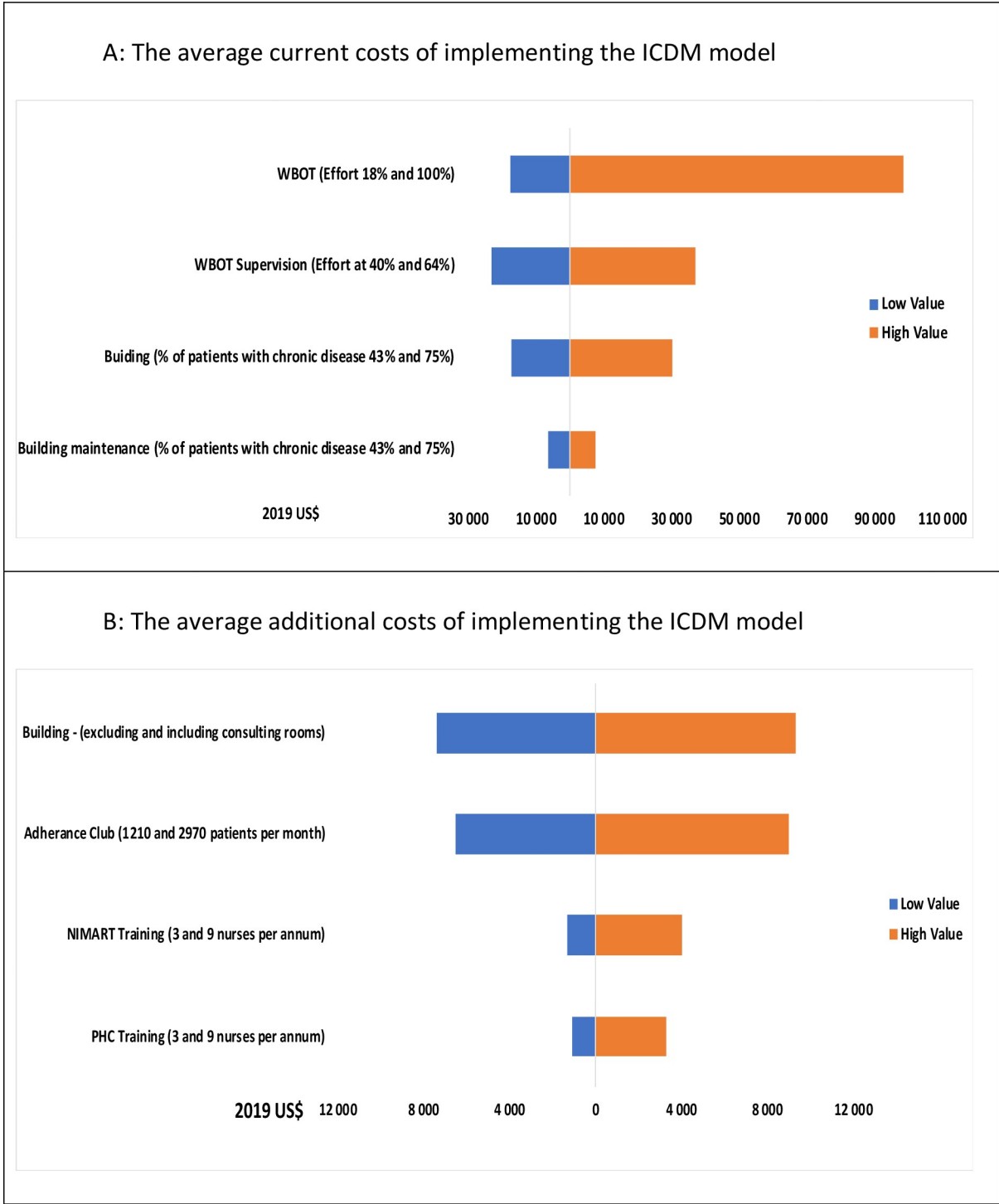

**Fig 1. Torando diagram summarizing one-way sensitivity analyses of mean and cost of implementing the recommended activities of the ICDM model per PHR clinic.** Middle line = zero costs.

reflects the increased allocation of financial resources by the South African Department of Health to PHC clinics to ensure that there is adequate infrastructure, medicines and supplies at PHC clinics [32].

The major contributor to the additional cost was additional infrastructure required for facility reorganization activities. These costs are lower than the cost of assisted self-management and clinical supportive management. However, the expenditure requires capital investment for which the health system might not have an available allocated budget. Clinical supportive management activities were the least costly, contributing 6% of the total cost. A report from South Africa has revealed that chronic diseases (diabetes, hypertension, asthma and epilepsy) are not adequately managed at PHC clinics, and there are few healthcare workers with adequate and competent training [37]. Our study findings support the view that a relatively low-cost investment in clinical supportive management can potentially have a high yield, improving quality of care for patients with chronic diseases.

Assisted self-management was the most costly component of the ICDM model to implement, and contributed 73% of the current cost and 31% of the additional cost. There was a great variability among clinics as some clinics were allocated more personnel. Human resource cost was also the highest cost in a team-based chronic care model [36]. In a study in Poland, the addition of integrated home-orientated services that included education and family support for patients with advanced chronic obstructive pulmonary disease (COPD) reduced the overall cost of care from the health system perspective [38]. The main reasons for cost reduction were reduced hospitalizations and disease exacerbations [38]. However, in other studies, the chronic care model reduced costs related to unplanned hospitalization, but increased costs for out-patient consultations and medication [39, 40]. Community-based adherence support and delivery of medication reduces the cost of accessing care from the patient's perspective. Cost has been mentioned as the greatest barrier to seeking care among 47% of patients in South Africa [41]. Affordability is greatly affected by travel cost as many of the PHC clinics are not within a walking distance of patients, especially in rural areas [42]. Despite the high cost of assisted self-management, it is an important component of chronic disease management to support adherence and reduce overcrowding at PHC clinics.

The cost of implementing the ICDM model activities varied across the four study PHC clinics, and the major cost drivers were the number of patients with chronic diseases consulted, as well as personnel for assisted self-management and the required additional infrastructure. High building costs for large clinics also contributed to some of the variation in clinic costs. A systematic review of chronic disease management programmes implemented in high-income countries showed that 14/16 (88%) of the studies demonstrated cost-effectiveness based on quality-adjusted life-year, while two studies had less than US$30 000 per life-year gained. Some of the reasons for the differences in the outcomes include the components of the chronic disease management programme that were implemented, the type of chronic disease being treated and level of comprehensiveness in measuring the costs [43]. In another study on a diabetes quality improvement project in five clinics in the USA, the cost of providing chronic care services per patient with diabetes per year varied from $6 in large clinics to $68 in small clinics [44].

## Strengths and limitations

The strength of this study is that it is one of the few studies that provides a cost analysis of implementing a chronic disease management model in a middle-income country. Moreover, most of the costs in the cost analysis are based on the actual costs for clinics to implement the ICDM model recommended activities. The results of this cost analysis could also be used for additional economic evaluations of the ICDM model. As data were collected from only four PHC clinics, a limitation of the study is that this small sample size could have resulted in large variability in the observed results. The generalizability of these results could also be limited by

this small sample size. However, we minimized this by including clinics with different degrees of implementation fidelity to the ICDM model from two different provinces. Therefore, this data would be informative for budgeting and resource allocation for clinics with similar characteristics. Another limitation of this analysis is that it does not include a full cost-effectiveness analysis as there is limited data on the effectiveness of the ICDM model in South Africa. The standard healthcare costs, like laboratory investigations, treatment of complications and medication was not included in the calculations and that leads to an underestimation of the overall cost of providing care to patients with chronic diseases. Lastly, only three of the four components of the ICDM model were included in the analysis. Therefore more research is needed on the implementation costs of the ICDM model activities related to strengthening of support systems.

## Conclusion

The estimated mean cost of implementing the ICDM model activities was $6.00 per patient with chronic disease per consultation. The greater portion of the costs are current costs. The additional cost of implementing the ICDM model with higher fidelity was minimal and comprised mainly of costs for facility reorganization and training of personnel. The mean cost of implementing the ICDM model activities varied between clinics and were affected by patient case load and the required additional infrastructure. The results of this cost analysis can enable additional ICDM model cost evaluations and budgetary planning for scale-up and scale-out of the ICDM model or similar models in countries with a similar disease burden and resources. Furthermore, this study provides information on the additional resources required to enhance the fidelity to the implementation of the ICDM model. Further research is needed on the cost-effectiveness of implementing the ICDM model for the management of patients with chronic diseases in South Africa and other similar contexts, as most of the studies published on cost-effectiveness of chronic care models are from high-income countries.

## Acknowledgments

We would like to acknowledge the personnel in the four primary healthcare clinics who allowed us to observe them at work and answered questions on the functioning of the clinics.

## Author Contributions

**Conceptualization:** Limakatso Lebina, Mary Kawonga, Tolu Oni, Olufunke A. Alaba.

**Data curation:** Limakatso Lebina.

**Formal analysis:** Limakatso Lebina, Hae-Young Kim, Olufunke A. Alaba.

**Funding acquisition:** Limakatso Lebina.

**Supervision:** Mary Kawonga, Tolu Oni, Olufunke A. Alaba.

**Writing – original draft:** Limakatso Lebina.

**Writing – review & editing:** Limakatso Lebina, Mary Kawonga, Tolu Oni, Hae-Young Kim, Olufunke A. Alaba.

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
