## [Decision Letter · Decision Letter 0]

18 May 2020

PONE-D-20-09044

The cost and cost implications of implementing the integrated chronic disease management model in South Africa

PLOS ONE

Dear Dr. Lebina,

Thank you for submitting your manuscript to PLOS ONE. After careful consideration, we feel that it has merit but does not fully meet PLOS ONE’s publication criteria as it currently stands. Therefore, we invite you to submit a revised version of the manuscript that addresses the points raised during the review process.

We would appreciate receiving your revised manuscript by June 15, 2020. To enhance the reproducibility of your results, we recommend that if applicable you deposit your laboratory protocols in protocols.io, where a protocol can be assigned its own identifier (DOI) such that it can be cited independently in the future. For instructions see: http://journals.plos.org/plosone/s/submission-guidelines#loc-laboratory-protocols

We look forward to receiving your revised manuscript.

Kind regards,

Alana T Brennan

Academic Editor

PLOS ONE

Journal Requirements:

2. Please include your tables as part of your main manuscript and remove the individual files. Please note that supplementary tables (should remain/ be uploaded) as separate "supporting information" files

3. Your ethics statement must appear in the Methods section of your manuscript. If your ethics statement is written in any section besides the Methods, please move it to the Methods section and delete it from any other section. Please also ensure that your ethics statement is included in your manuscript, as the ethics section of your online submission will not be published alongside your manuscript.

Reviewers' comments:

Reviewer's Responses to Questions

**Comments to the Author**

1. Is the manuscript technically sound, and do the data support the conclusions?

Reviewer #1: Yes

Reviewer #2: Yes

2. Has the statistical analysis been performed appropriately and rigorously? 

Reviewer #1: Yes

Reviewer #2: Yes

3. Have the authors made all data underlying the findings in their manuscript fully available?

Reviewer #1: Yes

Reviewer #2: Yes

4. Is the manuscript presented in an intelligible fashion and written in standard English?

Reviewer #1: Yes

Reviewer #2: Yes

5. Review Comments to the Author

Reviewer #1: Thanks for the chance to review this paper. The authors present cost estimates to implement an integrated model for the care of chronic diseases in primary health care clinics in two South African health districts to provide data to inform future implementation and evaluations of the program. The paper is well-written and I have only the following minor suggestions for the authors to consider:

Introduction

• Lines 98 – 100 – “Implementation costs is one of the implementation outcomes that also allows decision makers to determine and choose which interventions are effective, efficient and equitable” – I’m not sure this is accurate, e.g. how can the costs of implementation inform a policy decision around an interventions effectiveness?

• Lines 101-104 – “A process evaluation of the implementation fidelity of the ICDM model indicates that the level of fidelity to guidelines varies between clinics, with some clinics having high scores (80%-89%), and others having medium (70%-79%) and low (<70%) scores [19].” Should this be ‘adherence’ to guidelines?

Methods

• Lines 151-2 – I think the exchange rate is the wrong way around.

• Why are costs presented per patient visit? Would cost per patient be a better measure given the long term care needs of these patients? Is this possible with the data?

Discussion

• Line 301 – I think ‘healthy’ should be ‘health

Reviewer #2: PONE-D-20-09044

Lebina et al in their study on ‘The cost and cost implications of implementing the integrated chronic disease management model in South Africa’ undertake a cost analysis of implementing the integrated chronic disease management (ICDM) model in four primary healthcare clinics in South Africa, two with low and two with high fidelity.

The manuscript is well written and reports on the cost aspects of a larger study to assess the levels and determinants of fidelity in the implementation of the ICDM model in two health districts in South Africa. There are however, several shortcomings as denoted below that would need to be considered prior to publication of the study.

Abstract:

Methods: The abstract currently lacks information on the components of the ICDM evaluated – facility reorganisation, clinical, supportive management, assisted self-management. A sensitivity analysis is referred to, without further information for the reader to understand to what this analysis refers to.

Results: The overall costs of implementing the ICDM model are not very informative as generally the reader has no reference to the usual costs of providing services in the primary healthcare clinics included in the study. In this regard referring to the percentage of funds needed on top of the annual costs to run these clinics may be much more informative (also see lines 230-231); this would also take into account that clinics may differ with regard to size and patients demographics / disease profile.

Main manuscript

Introduction: Please clarify in line 58, that reference 9 refers to national and reference 10 to provincial figures.

Methods: Two of all the districts where the ICDM pilot was undertaken were included in the study. Please provide the rationale for the selection of these two districts. The costing undertaken in the current study has not considered standard healthcare service costs like medication, laboratory investigations and management of complications. Kindly provide the rationale why the costing of these factors has not been included and how this could have influenced the results of the study.

Results: In line with the note above relating to the abstract an increasing focus on the percentage of additional costs for implementation of the ICDM per clinic would be considerably more informative than the overall additional costs of the ICDM implementation. Further, inclusion of once-off capital investment costs in the overall cost estimate, although providing an overall cost estimate, may be at the same time less informative than differentiating between once-off capital investment costs for e.g. building infrastructure and recurring / annual costs to maintain these / offer services.

Limitations: Generally limitations are well discussed, however, given the extent of these limitations, could the authors discuss in more detail the implications of these shortcomings?

Ethics approval: Please add the ethics certificate number.

References: It seems that several conference presentations have been included in the reference list, please check for updates, respectively published articles such as Ameh etal: J Acquir Immune Defic Syndr. 2017 Aug 1;75(4):472-479. doi: 10.1097/QAI.0000000000001437.Effectiveness of an Integrated Approach to HIV and Hypertension Care in Rural South Africa: Controlled Interrupted Time-Series Analysis.

Please also check for completeness of references (e.g. ref 25) and remove duplicate references (e.g. ref 19 & 24).

Table 1: Kindly include further information on general characteristics and infrastructure / equipment of the clinics.

Table 2: For the reader table 2 is difficult to understand. What is meant by current and additional ICDM costs if the overall estimated costs of implementing the ICDM are the sum of the two? For clinic B (high fidelity) considerably higher building costs are included – these should be reported in the results and possibly explanations referred to in the discussion. For clinic C (low fidelity), a very high figure for WBOT assisted self-management is reported, please check the figure and if confirmed, report it in the result section and elaborate on possible explanations and implications of these study results (and summary figures presented) in the discussion section.

Figures: Figures 1a and 1b are of very poor quality not allowing to assess the content of these figures in detail.

General comment: The paper refers to developed and developing countries which in my understanding is rather an outdated terminology. I recommend to use the terms low- and middle-income, respectively high income countries / settings, especially in the case of South Africa considered to be an emerging economy.

6. PLOS authors have the option to publish the peer review history of their article (what does this mean?). If published, this will include your full peer review and any attached files.

Reviewer #1: No

Reviewer #2: No

---

## [Author Response · Author response to Decision Letter 0]

11 Jun 2020

11 June 2020

Dear Editor

Re: Manuscript: “The cost and cost implications of implementing the integrated chronic disease management model in South Africa.”

Manuscript ID: PONE-D-20-09044

Thank you very much for consideration of the above named manuscript for publication in your journal. 

We thank the reviewers for their comments. The manuscript has been reviewed and revised for clarity and accuracy.

Please see below for the itemised responses to reviewers’ comments.

Yours Sincerely

Limakatso Lebina 

On behalf of all authors.

 

*Line references are on tracked manuscript

Reviewer #1: 

Thanks for the chance to review this paper. The authors present cost estimates to implement an integrated model for the care of chronic diseases in primary health care clinics in two South African health districts to provide data to inform future implementation and evaluations of the program. The paper is well-written and I have only the following minor suggestions for the authors to consider:

Introduction

1. Lines 98 – 100 – “Implementation costs is one of the implementation outcomes that also allows decision makers to determine and choose which interventions are effective, efficient and equitable” – I’m not sure this is accurate, e.g. how can the costs of implementation inform a policy decision around an interventions effectiveness?

Thank you for this comment, we have revised the sentence for clarity.

Lines 117-119: “Implementation costs is one of the implementation outcomes that also allows decision makers to determine and choose which interventions are efficient and equitable [20].”

2. Lines 101-104 – “A process evaluation of the implementation fidelity of the ICDM model indicates that the level of fidelity to guidelines varies between clinics, with some clinics having high scores (80%-89%), and others having medium (70%-79%) and low (<70%) scores [19].” Should this be ‘adherence’ to guidelines?

Thank you for this comment. The sentence has been revised as suggested.

Lines 120-123: “A process evaluation of the implementation fidelity of the ICDM model indicates that the level of fidelity (adherence) to guidelines varies between clinics, with some clinics having high scores (80%-89%), and others having medium (70%-79%) and low (<70%) scores [24].”

Methods

3. Lines 151-2 – I think the exchange rate is the wrong way around.

Thank you for this comment. We have corrected the sentence.

Lines 176-177: “Costs were stated according to the 2019 prices and converted to US dollars (US$) at an exchange rate of ZAR 14.42 equal to US$ 1[30].”

4. Why are costs presented per patient visit? Would cost per patient be a better measure given the long term care needs of these patients? Is this possible with the data?

Thank you for this comment. Yes, it’s not possible with this routinely collected data on patient caseload to present costs per patient. 

Discussion

5. Line 301 – I think ‘healthy’ should be ‘health

Thank you for this comment. We have corrected the spelling error in that sentence. 

Lines 366-367: “However, the expenditure requires capital investment for which the health system might not have an available allocated budget.”

Reviewer #2: PONE-D-20-09044

Lebina et al in their study on ‘The cost and cost implications of implementing the integrated chronic disease management model in South Africa’ undertake a cost analysis of implementing the integrated chronic disease management (ICDM) model in four primary healthcare clinics in South Africa, two with low and two with high fidelity.

The manuscript is well written and reports on the cost aspects of a larger study to assess the levels and determinants of fidelity in the implementation of the ICDM model in two health districts in South Africa. There are however, several shortcomings as denoted below that would need to be considered prior to publication of the study.

Abstract:

1. Methods: The abstract currently lacks information on the components of the ICDM evaluated – facility reorganisation, clinical, supportive management, assisted self-management. A sensitivity analysis is referred to, without further information for the reader to understand to what this analysis refers to.

Thank you for this comments. We have revised the methods section of the abstract to include the additional information.

Lines 25-28: “We estimated the costs of implementing the ICDM model current activities for three (facility reorganization, clinical supportive management and assisted self-management) components and additional costs of implementing with enhanced fidelity.”

Lines 30-32: “One-way sensitivity analyses were carried out for key parameters by varying patient caseloads, required infrastructure and staff.”

2. Results: The overall costs of implementing the ICDM model are not very informative as generally the reader has no reference to the usual costs of providing services in the primary healthcare clinics included in the study. In this regard referring to the percentage of funds needed on top of the annual costs to run these clinics may be much more informative (also see lines 230-231); this would also take into account that clinics may differ with regard to size and patients demographics / disease profile.

Thank you for this comment. We have revised the results section of the abstract to indicate additional funds required in addition to what is already being spent on current ICDM model activities.

Lines 35-37: “Current ICDM model activities cost accounted for 84% ($124 345.00) of the annual mean cost, while additional costs for higher fidelity were 16% ($24 102.00).”

Main manuscript

3. Introduction: Please clarify in line 58, that reference 9 refers to national and reference 10 to provincial figures.

Thank you for this comment, we have revised the sentence to improve clarity.

Lines 70-73: “Data from Free State Province indicate that 24% of the population suffer from chronic diseases [9]. Results from a Cape Town study showed that prevalence of multi-morbidity is approximately 23%, and chronic diseases account for 45% of all primary healthcare (PHC) consultations [10].” 

4. Methods: Two of all the districts where the ICDM pilot was undertaken were included in the study. Please provide the rationale for the selection of these two districts. 

Thank you for this comment, the two health districts (out of three pilot health districts) were selected for this study as they have comparable population sizes and burden of disease. 

Lines 154-156: “The two health districts were selected for this study as they have comparable population sizes and burden of disease.”

5. The costing undertaken in the current study has not considered standard healthcare service costs like medication, laboratory investigations and management of complications. Kindly provide the rationale why the costing of these factors has not been included and how this could have influenced the results of the study.

Thank you for this comment, we have provided the rationale for not including the standard of care costs.

Lines 147-149: “Standard healthcare service costs like medication, laboratory investigations and management of complications were excluded from the analysis to keep the focus on the ICDM model activities. The standard costs would be incurred irrespective of the ICDM model and their inclusion would overestimate the costs of the ICDM model activities.” 

6. Results: In line with the note above relating to the abstract an increasing focus on the percentage of additional costs for implementation of the ICDM per clinic would be considerably more informative than the overall additional costs of the ICDM implementation. Further, inclusion of once-off capital investment costs in the overall cost estimate, although providing an overall cost estimate, may be at the same time less informative than differentiating between once-off capital investment costs for e.g. building infrastructure and recurring / annual costs to maintain these / offer services.

Thank you for this comment. The results section has been updated to increase the focus on proportion of additional costs to enhance fidelity and also estimated costs that excludes capital costs.

Lines 252-268: “The mean overall annual cost of implementing the recommended activities of the ICDM model per facility was $148 447.00 (SD: $65 125.00), and the mean cost per patient per visit was $6.00 (SD:$0.77). Capital costs were 24% ($35 760.09) of the total annual mean costs. Current cost contributed 84% ($124 345.00) of the total annual mean cost, while additional cost accounted for 16% ($24 102.00) (Table 3). Almost two-thirds of the current cost came from the personnel cost of the WBOTs/CHCW and their supervision under assisted self-management (66%; $91 204.00). For the additional annual cost, facility reorganization accounted for 49% ($9 668.64), while adherence clubs and the training cost for nurses were 31% ($7 470.80) and 20% ($4 868.60) respectively. The additional recurrent costs to achieve higher ICDM model fidelity per patient per visit was $0.62 (SD: 0.15).” 

7. Limitations: Generally limitations are well discussed, however, given the extent of these limitations, could the authors discuss in more detail the implications of these shortcomings?

Thank you for this comment. The limitations section has been updated to include more details regarding the implications of the shortcomings of the study.

Lines 412-427: “As data were collected from only four PHC clinics, a limitation of the study is that this small sample size could have resulted in large variability in the observed results. The generalizability of these results could also be limited by this small sample size. However, we minimized this by including clinics with different degrees of implementation fidelity to the ICDM model from two different provinces. Therefore, this data would be informative for budgeting and resource allocation for clinics with similar characteristics. Another limitation of this analysis is that does not include a full cost-effectiveness analysis as the there is no data on the effectiveness of the ICDM model in South Africa. The standard healthcare costs, like laboratory investigations, treatment of complications and medication was not included in the calculations and that leads to an underestimation of the overall cost of providing care to patients with chronic diseases. Lastly, only three of the four components of the ICDM model were included in the analysis. Therefore more research is needed on the implementation costs of the ICDM model activities related to strengthening of support systems.”

8. Ethics approval: Please add the ethics certificate number.

Thank you for this comment, we have included the ethics committees’ certificate numbers.

Line 165-166: “Ethics Approval: The study was approved by the ethics committees of the University of Cape Town (Ref: 127/2018) and the University of the Witwatersrand (Ref: R14/49).”

9. References: It seems that several conference presentations have been included in the reference list, please check for updates, respectively published articles such as Ameh etal: J Acquir Immune Defic Syndr. 2017 Aug 1;75(4):472-479. doi: 10.1097/QAI.0000000000001437.Effectiveness of an Integrated Approach to HIV and Hypertension Care in Rural South Africa: Controlled Interrupted Time-Series Analysis.

Please also check for completeness of references (e.g. ref 25) and remove duplicate 

references (e.g. ref 19 & 24).

Thank you for this comment. All references have been reviewed and updated as necessary.

10. Table 1: Kindly include further information on general characteristics and infrastructure / equipment of the clinics.

Thank you for this comment. Table 1 has been updated to include more characteristics of the PHC clinics. 

11. Table 2: For the reader table 2 is difficult to understand. What is meant by current and additional ICDM costs if the overall estimated costs of implementing the ICDM are the sum of the two? 

Thank you for this comment. We revised Table 2 headings and subheadings for clarity. Please see below.

12. For clinic B (high fidelity) considerably higher building costs are included – these should be reported in the results and possibly explanations referred to in the discussion. 

Thank you for this comment, the building costs for clinic B are high because of the large size of the clinic that is in a repurposed municipality building. We have updated the results and the discussion sections to include an explanation for this high costs.

Lines 282-283: “The building costs for clinic B are higher compared to the other clinics it is based in a large repurposed municipality building.” 

Line 398: “High building costs for large clinics also contributed to some of the variation in clinic costs.” 

13. For clinic C (low fidelity), a very high figure for WBOT assisted self-management is reported, please check the figure and if confirmed, report it in the result section and elaborate on possible explanations and implications of these study results (and summary figures presented) in the discussion section.

Thank you for this comment. The number of allocated WBOTs and CHCW is dependant 

Lines 305-308: “The costs per clinic for WBOTs/CHCW differed by clinic as it is greatly affected by the number of WBOTs/CHCW based at clinic according to the number of community wards. For example the cost for clinic C was $145 228.22 for 50 WBOTs/CHCW versus $11 618.25 in clinic A for 4 WBOTs/CHCW.” 

Lines 379-380: “There was a great variability among clinics as some clinics were allocated more personnel.”

14. Figures: Figures 1a and 1b are of very poor quality not allowing to assess the content of these figures in detail.

Thank you for this comment. We have made some changes to improve the quality of the Figures.

15. General comment: The paper refers to developed and developing countries which in my understanding is rather an outdated terminology. I recommend to use the terms low- and middle-income, respectively high income countries / settings, especially in the case of South Africa considered to be an emerging economy.

Thank you for this comment, we have revised the manuscript according to your recommendations. 

Lines 55-56: “Further research on the cost-effectiveness of the ICDM model in middle-income countries is required.”

 Lines 68-70: “South Africa, like many low and middle-income countries, has a dual burden of chronic diseases, with a high prevalence of both communicable and non-communicable chronic diseases [7, 8].” 

Lines 358-359: “All the studies that were reviewed had been conducted in high-income countries [16].” 

Lines 399-400: A systematic review of chronic disease management programmes implemented in high-income countries showed that 14/16 (88%) of the studies demonstrated cost-effectiveness.”

---

## [Editor Report · Decision Letter 1]

16 Jun 2020

The cost and cost implications of implementing the integrated chronic disease management model in South Africa

PONE-D-20-09044R1

Dear Dr. Lebina,

We’re pleased to inform you that your manuscript has been judged scientifically suitable for publication and will be formally accepted for publication once it meets all outstanding technical requirements.

Kind regards,

Alana T Brennan

Academic Editor

PLOS ONE
---

## [Editor Report · Acceptance letter]

17 Jun 2020

PONE-D-20-09044R1 

The cost and cost implications of implementing the integrated chronic disease management model in South Africa 

Dear Dr. Lebina:

I'm pleased to inform you that your manuscript has been deemed suitable for publication in PLOS ONE. Congratulations! Your manuscript is now with our production department. 

Kind regards, 

on behalf of

Dr. Alana T Brennan 

Academic Editor

PLOS ONE